# Abnormal p53 High-Grade Endometrioid Endometrial Cancer: A Systematic Review and Meta-Analysis

**DOI:** 10.3390/cancers17010038

**Published:** 2024-12-26

**Authors:** João Casanova, Alexandru Babiciu, Gonçalo S. Duarte, Ana Gomes da Costa, Sofia Silvério Serra, Teresa Costa, Ana Catarino, Mário M. Leitão, Jorge Lima

**Affiliations:** 1Gynecologic Oncology Unit, Obstetrics and Gynecology Service, Department of Surgery, Hospital da Luz Lisboa, 1500-650 Lisbon, Portugal; joao.miguel.casanova@hospitaldaluz.pt (J.C.); ana.g.costa@hospitaldaluz.pt (A.G.d.C.); 2Laboratory of Clinical Pharmacology and Therapeutics, Faculty of Medicine, University of Lisbon, 1649-004 Lisbon, Portugal; alexandrubabiciu@edu.ulisboa.pt (A.B.); gduarte@edu.ulisboa.pt (G.S.D.); 3Clinical Pharmacology Unit, Unidade Local de Saúde Santa Maria, 1649-035 Lisbon, Portugal; 4Library of NOVA Medical School, Universidade Nova de Lisboa, 1099-085 Lisbon, Portugal; sofia.serra@nms.unl.pt (S.S.S.); teresa.costa@nms.unl.pt (T.C.); 5Department of Pathology, Hospital da Luz Lisboa, 1500-650 Lisbon, Portugal; acatarino@hospitaldaluz.pt; 6Gynecology Service, Department of Surgery, Memorial Sloan Kettering Cancer Center, New York, NY 10065, USA; leitaom@mskcc.org; 7Gynecologic Service, Weil Cornell Medical College, New York, NY 10065, USA; 8Comprehensive Health Research Center (CHRC), NOVA Medical School, Faculdade de Ciências Médicas, NMS, FCM, Universidade NOVA de Lisboa, 1099-085 Lisbon, Portugal; 9Department of Obstetrics and Gynecology, Luz Saúde, Hospital da Luz Lisboa, 1500-650 Lisbon, Portugal

**Keywords:** endometrial cancer, high-grade endometrioid, p53, systematic review, meta-analysis, overall survival, progression-free survival, prevalence

## Abstract

Endometrial cancer is the most common gynecologic malignancy in the Western hemisphere. Grade 3 endometrioid endometrial cancer is a heterogeneous malignancy with different prognoses according to the stage and the molecular subtype described by the TCGA consortium. We found that women with abnormal p53 FIGO grade 3 endometrioid endometrial cancer have an increased risk of death (HR, 1.29; 95% CI, 1.11–1.48; I^2^ = 88%) and disease progression (HR, 1.63; 95% CI, 1.42–1.88; I^2^ = 2%) compared with women with wildtype p53 high-grade endometrioid carcinomas. The estimated pooled prevalence of abnormal p53 FIGO grade 3 endometrioid endometrial cancer is 30%. We also show that the prevalence of abnormal p53 in G3 endometrioid endometrial cancer has geographical variations, and, according to our study, the prevalence is higher in Asia.

## 1. Introduction

Endometrial cancer is the most common gynecologic malignancy in Western countries, and its incidence and mortality are rising [1]. According to the two-tier classification described by Bokhman in 1983, endometrial carcinomas are classified as type I (hormone-dependent) and type II (hormone-independent) tumors [2]. Type I tumors express high levels of estrogen receptors, and this excess estrogen, without the unopposed signaling of progesterone, plays a major role in the uncontrolled growth of the endometrium. The link between obesity, excess estrogen, and endometrial cancer derives from the fact that adipose tissue expresses aromatase, which converts androgens in estrogens, promoting the uncontrolled growth of the endometrial tissue in the absence of progesterone [3]. Type I tumors have an endometrioid histology and comprise approximately 80% of all endometrial cancer cases. Type II tumors rarely express estrogen, usually arise from an atrophic endometrium, and have a non-endometrioid histology, including serous, clear-cell, and carcinosarcoma morphologies [1,2]. However, this simplified approach to classifying endometrial carcinomas is no longer valid with the continued refinement of our pathologic and molecular understanding of these diseases.

Endometrioid endometrial carcinoma is graded according to the International Federation of Obstetrics and Gynecology (FIGO) system. This system uses a scale ranging from 1 to 3 and refers to the ratio of glandular to solid tumor elements [4]. Grade 1 (G1) and grade 2 (G2) tumors, or low-grade tumors, usually have a favorable prognosis. Grade 3 (G3) tumors (high-grade tumors) are associated with diverse prognoses, sometimes resembling those of non-endometrioid carcinomas. Thus, G3 endometrioid tumors can be considered somewhat of a clinical and pathological dilemma. Clinically, the natural history of these carcinomas can mimic the most aggressive non-endometrioid histologies or have a relatively favorable prognosis resembling that of low-grade endometrioid endometrial carcinomas [5]. It is important to underline that the histological diagnosis is also controversial as it is associated with interobserver variability and poor reproducibility [5].

Since the landmark publication by the Cancer Genome Atlas (TCGA) Research Network in 2013, the previously described two-tier classification has steadily been replaced by a molecular classification encompassing four distinct subgroups [6]. These molecular subtypes of endometrial carcinoma include copy number high (CN-H); copy number low (CN-L); an ultramutated group, which is defined by mutations in the exonuclease domain of the polymerase epsilon (*POLE*) gene; and a microsatellite-unstable subgroup (MSI) [6]. The TCGA involves significant molecular and genetic analyses, which are not possible in routine practice. Therefore, multiple other classifier systems have been proposed [7,8,9]. These somewhat mirror the TCGA groupings but are not exact and have led to group designations of *POLE*, a deficient mismatch repair (dMMR), no specific molecular profile (NSMP), and abnormal p53 [7,8,9].

Data from the literature show that G3 endometrioid endometrial carcinomas with different molecular signatures behave heterogeneously, with patients who have *POLE*-ultramutated tumors showing a survival advantage [10,11]. On the opposite side of the spectrum is abnormal p53 cases, with robust data showing detrimental oncologic outcomes [5,12,13,14,15,16]. Abnormal p53 has been defined as either an abnormal immunohistochemical staining pattern for p53 or the presence of a *TP53* mutation in next-generation sequencing.

Although abnormal p53 has been consistently associated with a poor prognosis in patients with endometrial cancer [17,18,19,20,21,22,23,24], there are limited data and small series specifically addressing the association between p53 status and outcomes in G3 endometrioid endometrial carcinomas. Thus, the primary objective of this systematic review and meta-analysis was to collate and summarize the available data on the oncologic outcomes (including overall survival (OS) and progression-free survival (PFS)) of patients with abnormal p53 G3 endometrioid endometrial carcinoma. We also sought to determine the global prevalence of abnormal p53 in this subtype of endometrial carcinoma and determine whether there are geographical variations in this prevalence.

## 2. Methods

### 2.1. Protocol and Registration

This systematic review and meta-analysis were planned before the online search (based on a prespecified population, outcome measures, study eligibility criteria, and statistical analyses). It was conducted according to the Meta-Analysis of Observational Studies in Epidemiology (MOOSE) [25] and the Preferred Reporting Items for Systematic Reviews and Meta-Analysis (PRISMA) [26] statement (see Appendix A. PRISMA checklist), and it was preregistered with PROSPERO (no: CRD42023495192) [27]. As systematic reviews and meta-analyses only involve the use of previously published data, no formal ethical approval or informed consent was required.

### 2.2. Eligibility Criteria

We included all references based on prespecified population, intervention, comparison, and outcome criteria as follows:Population: Women of reproductive age diagnosed with G3 endometrioid endometrial carcinoma with known p53 status using either genetic sequencing or an immunohistochemistry surrogate. Intervention: no intervention.Exposure: Women with abnormal p53 status diagnosed with G3 endometrioid endometrial carcinoma.Comparison: Wildtype p53 G3 endometrioid endometrial carcinoma.Primary outcomes: OS and PFS.Secondary outcomes: Global prevalence and geographical differences in the prevalence of p53-mutated G3 endometrioid EC. We also determined whether the year of publication, median patient age, proportion of patients with FIGO stage III or IV, and overall risk of bias had an impact on the prevalence estimates.

We only included English manuscripts in which p53 status was tested using genetic sequencing or immunohistochemistry and manuscripts clearly stating this information.

Other inclusion criteria included human studies; studies involving adequate clinical and pathological data; studies involving only high-grade endometroid endometrial carcinomas; clear information on oncologic outcomes (PFS and OS); and sufficient data allowing for calculations of the hazard ratio (HR), standard error (SE), and odds ratio (OR). The exclusion criteria were published abstracts without a published manuscript; single case reports, commentaries, letters to editors, editorials, and review articles; articles without enough data for calculation, without confirmation of p53 status using genetic sequencing or immunohistochemistry, or with inconclusive data regarding either histology or tumor grading (wrong population); and duplicate studies.

### 2.3. Information Sources and Search Strategy

A comprehensive review of the literature was performed on 8 January 2024. The literature search was performed using the major electronic databases available: PubMed/Medline, EMBASE, Cochrane Library, Scopus, and Web of Science. The search strategy (Appendix A) combined Boolean operators with the following search terms:Endometrial cancer OR endometrial carcinoma OR EC; high-grade endometrioid endometrial cancer OR G3 endometrioid endometrial cancer OR G3 endometrioid endometrial carcinoma; p53 mutant OR p53 mutation OR p53-mutated OR p53 abnormal OR p53-positive OR p53 positivity OR p53 immunohistochemistry OR *TP53* mutation.

No language, year of publication, or study type restrictions were applied to the search. References of the most relevant studies and reviews were also hand-screened to identify any eventual missing publications not retrieved by the electronic search. New searches were reperformed to ensure the inclusion of any eligible new publications during the conduction of this review. Artificial intelligence software was used to store, organize, and manage all references obtained from the literature search [28].

### 2.4. Study Selection, Data Extraction, and Data Items

Two reviewers (J.C. and A.B.) independently assessed all titles and abstracts of the retrieved search articles. The selection of full-text articles for inclusion was independently performed by two reviewers (J.C. and A.B.), and any disagreement was solved by a third independent reviewer (J.L.). All studies were independently analyzed by two reviewers (A.B. and A.G.C.), and disagreements were solved by a third independent reviewer (J.C.). Data were extracted by two reviewers (J.C. and A.G.C.) and evaluated by an additional reviewer (G.S.D.). Where applicable, the corresponding authors of the included studies were contacted to obtain or confirm data. Data on study authors, year of publication, study type, endometrial cancer cohort size, G3 cohort size, and abnormal p53 G3 endometrial cancer cohort size were extracted. Demographic data (including age), clinical data (including body mass index and follow-up duration), and pathological data (including FIGO stage and adjuvant therapy) of the study population were extracted, as well as the methods used to determine p53 status (sequencing method and/or immunohistochemistry). Data on OS, PFS, and the prevalence of abnormal p53 in G3 endometrioid endometrial carcinoma were also collected.

### 2.5. Assessment of Risk of Bias

One reviewer (G.S.D.) independently assessed the quality of the studies and the risk of bias using the Quality in Prognosis Studies (QUIPS) tool, as recommended by the Cochrane Prognosis Methods Group [29]. A second reviewer (J.C.) reviewed this assessment, and disagreements were solved by a third independent reviewer (J.L.). The QUIPS tool includes the following six domains to evaluate the validity and bias in studies of prognostic factors: study participation, study attrition, prognostic factor measurement, outcome measurement, study confounding, and study analysis and reporting [29]. The risk of bias was further categorized as high, intermediate, or low [29].

### 2.6. Data Synthesis and Statistical Analyses

For time-to-event data, we used the generic inverse variance method and pooled HR with corresponding 95% confidence intervals (CIs). For each study, we used individual patient data (IPD) if available from the study team. If IPD were not available, we extracted information about time-to-event outcomes using the methods previously described in the literature [30]. If studies were considered to be similar enough (in terms of patients, settings, intervention, and outcome measures) to allow for the pooling of data using a meta-analysis, we assessed the degree of heterogeneity by visually inspecting forest plots, estimating the percentage of heterogeneity between studies (which could not be ascribed to sampling variation, namely, I^2^), and, when possible, performing subgroup analyses.

We estimated patient-level survival data from published Kaplan–Meier curves using validated algorithms [30]. We downloaded, preprocessed, and digitized raster images of survivor curves to obtain their step function, including the step timings, using, if available, additional information such as number-at-risk tables and the total number of events to further improve the calibration of the reconstruction algorithm. Then, we recovered the time-to-event information for individual women by solving the inverted Kaplan–Meier product limit equations. Comparisons of the reconstructed and original Kaplan–Meier curves demonstrated that the algorithms robustly recovered patient-level survival times from the published studies.

We analyzed OS and PFS using both a one-stage method, as previously described [30] (i.e., using reconstructed or original individual patient data), and a two-stage approach (i.e., prespecified inverse variance-weighted meta-analyses). For the one-stage meta-analyses, we used the Kaplan–Meier method to calculate the OS. We carried out the one-stage meta-analyses using Cox proportional hazards models, which address between-study heterogeneity using a variety of approaches. We regarded the shared frailty model to be the most robust approach, as it most explicitly incorporates a gamma-distributed random-effects term to account for between-study heterogeneity. The median follow-up was calculated using the reverse Kaplan–Meier method. Post hoc sensitivity analyses were conducted for OS and PFS by only including data from trials using the reported aggregate-level data.

For prevalence calculation, the total number of individuals screened was used as the denominator. Data were subjected to Freeman–Tukey transformation (double arcsine transformation) to avoid negative prevalence in the CI, limiting the CI to between 0% and 100%. We pooled data using the empirical Bayes estimator for tau^2^. We conducted a subgroup analysis to explore the geographical variations in the prevalence of abnormal p53 in endometrioid EC, categorizing the studies according to their continent of origin (America, Europe, or Asia). We also conducted a regression analysis to determine whether the covariates’ year of publication, patient age (median), proportion of patients with FIGO stage III or IV, and overall risk of bias had an impact on the pooled prevalence estimate. Publication bias was assessed by conducting a linear regression of funnel plot asymmetry using Egger’s test. In the case of statistically significant evidence of publication bias using Egger’s test, we further conducted a confirmatory analysis using p-curve and trim-and-fill analyses. *p* < 0.05 was considered statistically significant. R statistical software (version 4.3.2) was used for all statistical analyses.

## 3. Results

### 3.1. Study Selection

The search yielded 3852 records. After removing 354 duplicates, 3498 were screened. After the screening process, 3360 records were excluded. Of the 138 reports sought for retrieval, 5 could not be retrieved and were excluded. A total of 133 reports were assessed for eligibility, and ultimately, 57 studies were included [5,7,9,13,14,15,17,18,19,20,21,22,23,24,31,32,33,34,35,36,37,38,39,40,41,42,43,44,45,46,47,48,49,50,51,52,53,54,55,56,57,58,59,60,61,62,63,64,65,66,67,68,69,70,71,72,73]. The full texts of 57 articles were assessed for eligibility; they met all of the inclusion criteria and were subsequently included in the analyses. The included articles were published between 1991 and 2024 and included a total of 2528 patients with G3 endometrioid endometrial carcinoma. A flowchart of the reference selection process is presented in Figure 1.

### 3.2. Study Characteristics

The characteristics of the included studies are presented in Table 1. The 57 studies included a total of 2528 patients with abnormal p53 G3 endometrioid endometrial carcinoma. Of the 57 studies, 6 [5,13,35,37,38,64] with OS as an endpoint and 8 [5,7,15,35,37,38,59,64] with PFS as an endpoint had pooled data and were analyzed (Table 1).

### 3.3. Risk of Bias of Included Studies

Of the 57 studies, 45 [5,13,14,15,17,18,19,23,31,33,34,35,36,37,38,39,40,42,43,44,45,46,47,48,49,50,51,52,53,54,55,57,58,59,60,61,62,63,64,65,66,67,68,72,73] were rated as having a low overall risk of bias, and 12 [7,9,20,21,22,24,32,41,56,69,70,71] were rated as having a high overall risk of bias (Table 2).

### 3.4. Synthesis of Results

#### 3.4.1. Overall Survival

We pooled aggregate-level data from two trials and reconstructed approximate IPD-level data from four trials, including a total of 1524 patients. Overall, we found a higher risk of death in patients with an abnormal p53 status than in patients with a wildtype p53 status (HR: 1.29; 95% CI: 1.11–1.48; I^2^ = 88%; six studies [5,13,35,37,38,64]). Due to fewer than 10 trials contributing data, we did not conduct a linear regression of funnel plot asymmetry using Egger’s test (Figure 2).

A sensitivity analysis of OS showed stable results with the same direction and magnitude of the pooled estimates compared with the primary analyses (HR: 1.10; 95% CI: 0.86–1.40; I^2^ = 44%; two studies [5,13]) (Appendix A).

Although FIGO staging is a determinant prognostic factor for OS, we could not stratify our data according to this staging system due to the absence of complete data regarding the stage of every G3 endometrial carcinoma case in the included studies.

#### 3.4.2. Progression-Free Survival

We pooled aggregate-level data from three trials and reconstructed approximate IPD-level data from five trials, including a total of 1595 patients. Overall, we found a higher risk of disease progression in patients with an abnormal p53 status than in patients with a wildtype p53 status (HR: 1.63; 95% CI: 1.42–1.88; I^2^ = 2%; eight studies [5,7,15,35,37,38,59,64]). Due to fewer than 10 trials contributing data, we did not conduct linear regression of funnel plot asymmetry using Egger’s test (Figure 3).

A sensitivity analysis of PFS showed stable results with the same direction and magnitude of the pooled estimates compared with the primary analyses (HR: 2.01; 95% CI: 1.40–2.90; I^2^ = 0%; three studies [5,7,15]) (Appendix A).

As for OS, we could not stratify the PFS data according to the FIGO staging due to the absence of complete data regarding the stage of every G3 endometrial carcinoma case in the included studies.

#### 3.4.3. Prevalence of Abnormal p53 (Sequencing and Immunohistochemistry) and Geographical Variations

We pooled data from 57 trials [5,7,9,13,14,15,17,18,19,20,21,22,23,24,31,32,33,34,35,36,37,38,39,40,41,42,43,44,45,46,47,48,49,50,51,52,53,54,55,56,57,58,59,60,61,62,63,64,65,66,67,68,69,70,71,72,73] with a total of 2528 patients. We calculated a global prevalence of abnormal p53 of 30% (95% CI: 25–34%; tau^2^ = 0.02; I^2^ = 74%; 57 studies [5,7,9,13,14,15,17,18,19,20,21,22,23,24,31,32,33,34,35,36,37,38,39,40,41,42,43,44,45,46,47,48,49,50,51,52,53,54,55,56,57,58,59,60,61,62,63,64,65,66,67,68,69,70,71,72,73]) (Figure 4). We also conducted subgroup analyses according to the continent where the studies were conducted, and we found statistically significant differences between the subgroups (*p* < 0.01). For America, we estimated a prevalence of 21% (95% CI: 17–25%; tau^2^ = 0; I^2^ = 0%; 11 studies [7,15,31,39,42,45,47,57,60,65,69]). For Europe, we estimated a prevalence of 30% (95% CI: 21–37%; tau^2^ = 0.03; I^2^ = 85%; 22 studies [5,9,14,20,23,32,33,34,35,36,40,43,46,52,53,58,59,61,62,64,66,73]). For Asia, we estimated a prevalence of 34% (95% CI: 27–41%; tau^2^ = 0.01; I^2^ = 52%; 24 studies [13,17,18,19,21,22,24,37,38,41,44,48,49,50,51,54,55,56,63,67,68,70,71,72]) (Figure 4).

We conducted a linear regression of funnel plot asymmetry using Peters’ test, which was suggestive of publication bias (*p* = 0.008). To further explore the possibility of publication bias, we conducted a p-curve analysis, which corroborated the previous analysis, thereby confirming the presence of small-study effects (Appendix A). We further conducted a trim-and-fill analysis, which can be interpreted as an approximate bias adjustment method. The trim-and-fill analysis resulted in an estimated pooled prevalence of 21% (95% CI: 16–26%; tau^2^ = 0.05; I^2^ = 82%; 57 studies [5,7,9,13,14,15,17,18,19,20,21,22,23,24,31,32,33,34,35,36,37,38,39,40,41,42,43,44,45,46,47,48,49,50,51,52,53,54,55,56,57,58,59,60,61,62,63,64,65,66,67,68,69,70,71,72,73]) (Appendix A). To further explore the data, we conducted meta-regression analyses according to the following factors: year of publication, patient age, FIGO stage, and overall risk of bias. Overall, we found a statistically significant effect only for the covariate year of publication (*p* < 0.0001) (Figure 5 and Appendix A).

## 4. Discussion

We found an increased risk of death and disease progression in patients with abnormal p53 G3 endometrioid endometrial carcinoma compared with their wildtype (or normal) counterparts. Furthermore, in a pooled analysis of 57 studies (with a total of 2528 patients), we found a global prevalence of 30% of abnormal p53 in G3 endometrioid endometrial carcinoma. After refining the prevalence by continent, we found a greater prevalence of p53-mutated G3 endometrioid endometrial cancer in Asia (34%) than in Europe (30%) and America (21%).

As previously described, the p53 onco-suppressor protein is a transcription factor that inhibits cell division or survival in response to various stresses, thus acting as a “gatekeeper” mechanism of cellular anticancer defenses [74,75]. *TP53* mutations result in a dysfunctional p53 protein, disrupting its tumor-suppressive functions and promoting tumorogenesis [75]. Bearing this in mind, in G3 endometrioid endometrial carcinoma with mutated *TP53*, aberrant signaling pathways contribute to uncontrolled cell proliferation, genomic instability, and resistance to apoptosis. These molecular alterations drive the aggressive phenotype observed in this subtype of endometrial cancer [13,64].

Considering the heterogeneity of G3 endometrioid endometrial carcinoma, there is a need to develop more tailored approaches [5]. The results of this review suggest that, in the presence of abnormal p53 G3 endometrioid endometrial carcinoma, patients may require different therapeutic options according to the disease stage at presentation [76]. Our review corroborates and strengthens the findings of the existing body of literature that abnormal p53 G3 endometrioid endometrial carcinomas have a poor prognosis with an increased risk of recurrence and decreased OS [5,6,12,31].

Geographical differences in the incidence and outcomes of endometrial carcinoma have been addressed in the literature [77]. We noted an increased prevalence of abnormal p53 in Asia. Asians more commonly have non-endometrioid histologies, corresponding to the copy number high/abnormal p53 subgroup of the TCGA classification [77]. However, despite this, Asians have consistently shown better oncologic outcomes than Caucasians, probably due to an earlier age at diagnosis (mean of 58.4 years compared with 65.1 years) [77,78]. A more refined analysis, focusing on type I endometrial carcinoma, also highlighted a difference between US- and foreign-born Asians [78]. This study found that US-born Asians had a significantly higher number of type I cancers [78], which may be due to their increased obesity and increased prevalence of diabetes. Zhang et al. [77] also found subtle differences concerning country of origin and highlighted that Japanese women had the lowest proportion of type I cancer. Considering that we included several Japanese cohorts in our review, we can argue that the increased prevalence of p53 mutations found in G3 endometrial carcinoma corroborates this fact. Data on regional differences in endometrial cancer are evolving, and there is a need to adjust studies to the molecular classification of endometrial carcinomas [79].

Focusing on the differences concerning the date of publication, we found that a *TP53* mutational status has always been a subject of interest in the literature on gynecologic oncology. However, since the landmark publication of the TCGA consortium, the number of research papers focusing on the different molecular subtypes has significantly increased [6]. We presume that the difference found in prevalence in the early publications compared with the most recent ones may be due to several factors. First, most of the early studies were conducted in Asia, and, as we previously explained, the data show that Asians more commonly have non-endometrioid histologies and p53-mutated tumors [77,78,79]. Second, the laboratory techniques are currently more refined, and thus, the prevalence found in later studies is expected to be more accurate. Third, our findings suggest that there may have been publication bias in the early studies, where results on more aggressive tumors or tumors with a poor prognosis were more likely to be published.

Our study has several strengths. First, by including a considerable number of studies and patients and by using studies from different populations, the generalizability of the results was increased. Our results regarding the pooled estimates of both OS and PFS with aggregate-level data and reconstructed approximate IPD were highly consistent. The sensitivity analyses showed stable results with the same direction and magnitude of the pooled estimates compared with the primary analyses. Finally, the strength of this meta-analysis is increased by the overall good quality of the individual studies included. With this review, we were able to summarize the available data regarding the more unfavorable outcomes of patients with G3 endometrioid endometrial carcinomas with abnormal p53. We also calculated different prevalences according to geographical area. Together, our findings shed additional light and provide a better understanding of this highly heterogenous malignancy.

Regarding the main limitations of this review, we acknowledge that our conclusions are not a “novelty”. As discussed previously, there are robust data in the literature associating p53 mutations/aberrations with a poor prognosis of endometrial cancer [35,37,80,81]. Nevertheless, together with other reviews, we provide a more in-depth picture of the outcomes of different molecular subtypes of G3 endometrioid endometrial carcinomas [11]. Another major limitation is that we analyzed all tumors based only on p53 status, without accounting for FIGO staging, which is a determinant prognostic factor for OS and PFS [76]. However, we could not stratify our data according to the FIGO stage due to the absence of complete data regarding the stage of the high-grade endometrioid carcinomas in the included studies. Another limitation of this review is the relatively low number of studies included that calculated the pooled OS and PFS. Furthermore, some of the included studies assessed p53 status only through DNA sequencing, without specifying the method used. Nevertheless, a high concordance between immunohistochemistry and sequencing results has been reported, mainly when considering next-generation sequencing [73].

## 5. Conclusions

Despite all of the advancements recently made in the molecular profiling of endometrial cancer, G3 endometrioid endometrial carcinoma remains, in some respects, a clinical dilemma [5,6]. Evidence suggests that tumors with *POLE* mutations behave more similarly to their low-grade counterparts, whereas abnormal p53 tumors have a phenotype resembling non-endometrioid histologies [11,73]. Refining the molecular subtypes of G3 endometrioid endometrial carcinomas is of utmost importance to develop more tailored adjuvant therapies, with more aggressive treatment being used for the most aggressive subtypes. While waiting for the results of ongoing trials, such as the PORTEC4a, clinicians need to make decisions that incorporate the molecular classification in daily practice [76,82]. Our results are in accordance with those in the literature, highlighting that abnormal p53 G3 endometrioid endometrial carcinomas have a higher risk of death and disease progression. Understanding that the clinical behavior of these tumors is significantly worse may help guide clinicians in tailoring their decisions regarding treatment strategies.

## Figures and Tables

**Figure 1 cancers-17-00038-f001:**
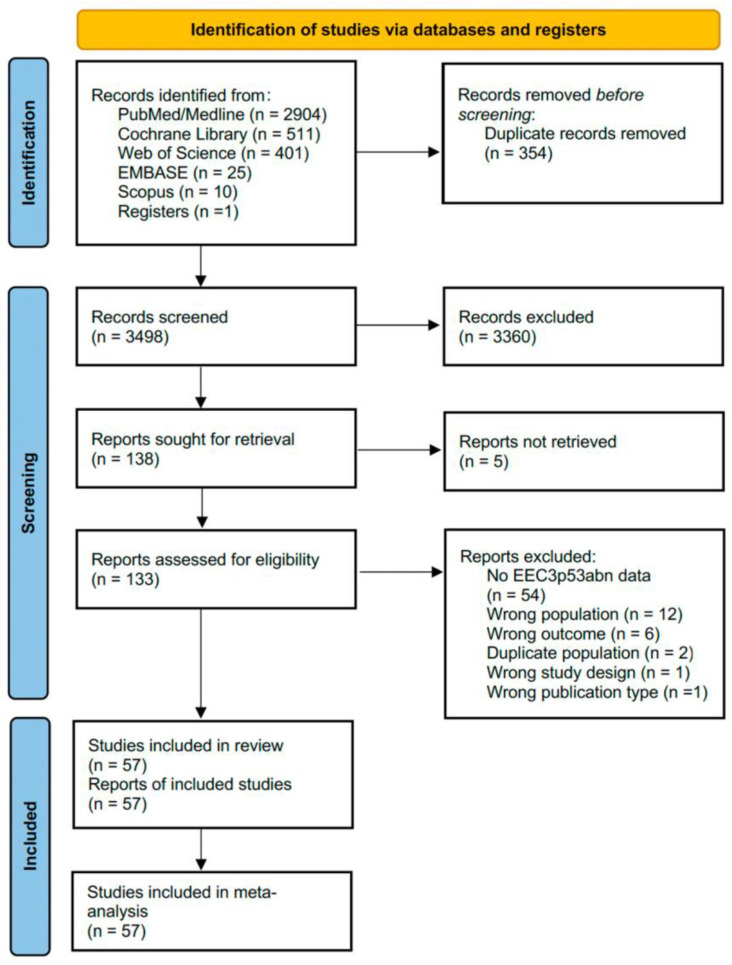
PRISMA 2020 flowchart of systematic review process and study selection. abn, abnormal; EEC, endometrioid endometrial cancer.

**Figure 2 cancers-17-00038-f002:**
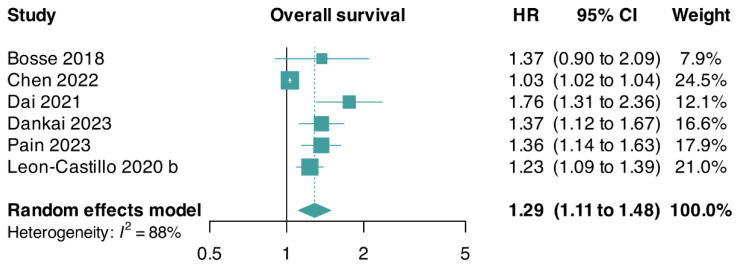
Forest plot for overall survival of patients with abnormal p53 G3 endometrioid endometrial carcinomas. CI, confidence interval; HR, hazard ratio [5,13,35,37,38,64].

**Figure 3 cancers-17-00038-f003:**
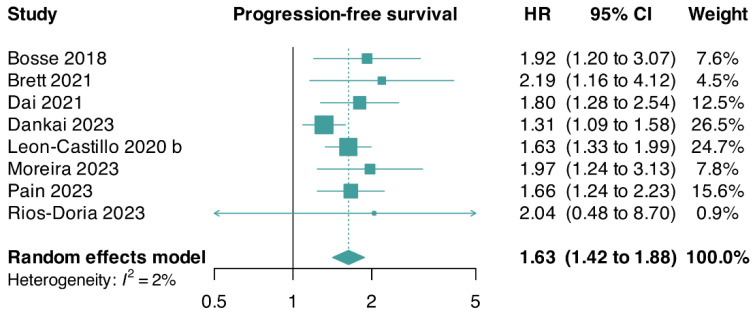
Forest plot for progression-free survival of patients with abnormal p53 G3 endometrioid endometrial carcinomas. CI, confidence interval; HR, hazard ratio [5,7,15,35,37,38,59,64].

**Figure 4 cancers-17-00038-f004:**
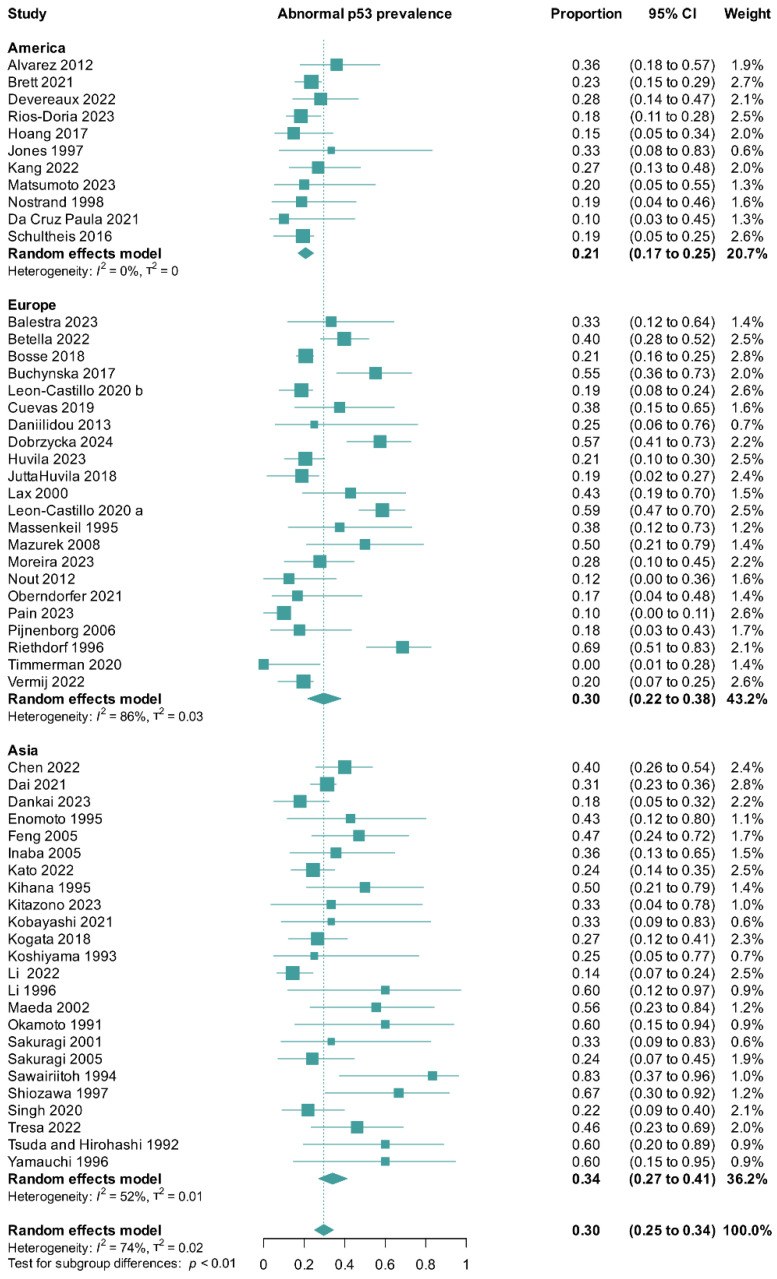
Global prevalence and geographical variation in prevalence of abnormal p53 in G3 endometrioid endometrial cancer. CI, confidence interval [5,7,9,13,14,15,17,18,19,20,21,22,23,24,31,32,33,34,35,36,37,38,39,40,41,42,43,44,45,46,47,48,49,50,51,52,53,54,55,56,57,58,59,60,61,62,63,64,65,66,67,68,69,70,71,72,73].

**Figure 5 cancers-17-00038-f005:**
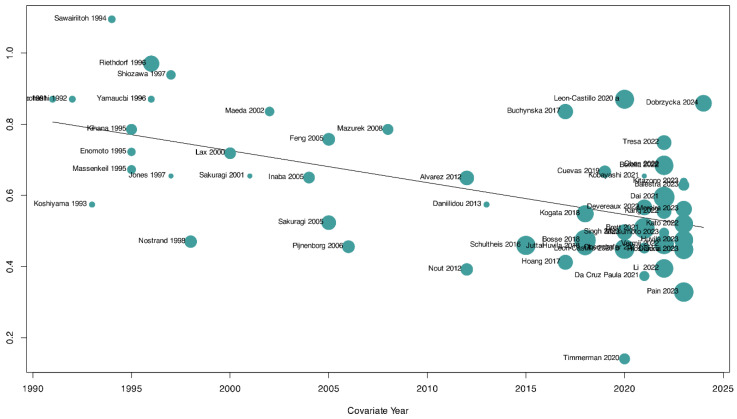
Meta-regression analysis of year of publication. The size of each circle is proportional to the total number of patients with abnormal p53 grade 3 endometrioid endometrial cancer included in each study. The *y*-axis corresponds to the estimated frequency of abnormal p53 status.

**Table 1 cancers-17-00038-t001:** The characteristics of the included studies.

Study, Year of Publication	Study Type	Country of Study	EEC Cohort Size	G3 Cohort Size	p53-Mutated G3 EEC Cohort Size	Age (y)	BMI (kg/m^2^)	FIGO Stage n/n(%)	Adjuvant Therapy n/n(%)/%	Method of Assessment of p53 Status	Follow-Up Duration	Has Kaplan–Meier Plot?	PFS ^a^ (HR [95% CI])	OS ^a^ (HR [95% CI])
Alvarez et al. [31]	Retrospective cohort study	USA	25	25	9	61 ^b^ (range: 37–88)	NS	I and II: 4 III–IV: 5	NS	IHC	6–96 mo	No	No	No
Balestra et al. [32]	Retrospective cohort study	Belgium	118	12	4	NS	NS	NS	NS	IHC, NGS	NS	No	No	No
Betella et al. [33]	Retrospective cohort study	Italy	278	73	29	65.0 ^b^ (SD, 10.9)	27.5	I to IV	NS	IHC	NS	No	No	No
Bosse et al. [5]	Retrospective cohort study	International	381	381	79	66 ^c^ (33–96)	NS	IA: 171 (45.5%) IB: 120 (31.9%) II–IV: 85 (22.6%)	Irrespective of adjuvant therapy given	IHC, Sanger sequencing, NGS	6.1 ^c^ (range 0.2–17) y	Yes	1.92 (1.20 to 3.07)	1.37 (0.90 to 2.09)
Brett et al. [15]	Prospective cohort study	Canada	326	200	47	65.07 ^b^ (SD, 12.04)	NS	I: 25 II: 4 III–IV: 18	RT: 16 CT: 24	IHC	54.61 ^c^ (0.61–151.20) mo	Yes	2.19 (1.16 to 4.12)	No
Buchynska et al. [34]	Retrospective cohort study	Ukraine	95	29	16	58.3 ^b^ (range: 36–83)	NS	I and II	NS	IHC	100 mo	No	No	No
Chen et al. [13]	Retrospective cohort study	China	60	60	24	56.22 ^b^ (SD, 10.51)	NS	NS	NS	NGS	NS	No	No	1.03 (1.02 to 1.04)
Cuevas et al. [36]	Retrospective cohort study	Spain	24	16	6	81.04 ^b^ (range: 59–96)	NS	IA: 4 (16.7%) IB: 14 (58.3%) II: 3 (12.5%) IIIA: 2 (8.3%) IIIB: 1 (4.2%)	NS	IHC, NGS	76 ^b^ (range: 24–108) mo	No	No	No
Da Cruz Paula et al. [65]	Retrospective cohort study	USA	175	10	1	62 ^c^ (range: 27–93)	NS	I: 129 (73.7%) II: 6 (3.4%) III: 30 (17.1%) IV: 10 (5.7%)	NS	IHC	32.3 ^c^ (range 1.3–94) mo	No	No	No
Dai et al. [37]	Retrospective cohort study	China	473	268	84	>65 y: 65.5%	≥28: 72.3%	III or IV: 41.1%	RT: 47.6% CT: 57.0%	NGS	30.6 ^c^ mo	Yes	1.80 (1.28 to 2.54)	1.76 (1.31 to 2.36)
Daniilidou et al. [14]	Retrospective cohort study	Greece	61	4	1	62.5 ^b^ (range: 39–75)	NS	NS	NS	IHC	5 y	No	No	No
Dankai et al. [38]	Retrospective cohort study	Thailand	138	39	7	57.2 ^b^ (range: 25–81)	NS	NS	NS	IHC	>60 mo	Yes	1.31 (1.09 to 1.58)	1.37 (1.12 to 1.67)
Devereaux et al. [39]	Prospective cohort study	USA	310	32	9	69 ^c^ (range: 34–87)	NS	I-IV	NS	IHC	20 mo	No	No	No
Dobrzycka et al. [40]	Prospective cohort study	Poland	139	40	23	<60 y: 58 ^d^ (41.7%) >60 y: 81 ^d^ (58.3%)	NS	IA: 9 (6.5%) IB: 58 (41.7%) IC: 72 (51.8%)	None	IHC, PCR	60 mo	Yes	No	No
Enomoto et al. [22]	Retrospective cohort study	Japan	38	7	3	NS	NS	NS	NS	DNA sequencing	NS	No	No	No
Feng et al. [41]	Retrospective cohort study	Japan	54	17	8	Range: 30–76	NS	p53-mutated: I + II: 15 III + IV: 8	NS	IHC, PCR	NS	No	No	No
Hoang et al. [42]	Retrospective cohort study	USA	115	27	4	NS	NS	NA	NS	IHC	NS	No	No	No
Huvila et al. [46]	Retrospective cohort study	Finland	306	53	10	66 ^c^ (range: 59–73)	NS	I: 247 (80.7%) II: 9 (2.9%) III: 42 (13.7%) IV: 8 (2.6%)	NS	Tissue microarrays, IHC	12 y	Yes	No	No
Huvila et al. [43]	Retrospective cohort study	Finland	957	78	16	≤60 y: 31.7% >60 y: 68.3%	NS	NS	NS	IHC, NGS	>5 y	Yes	No	No
Inaba et al. [44]	Retrospective cohort study	Japan	92	14	5	<60: 61 ^d^; ≥60: 31 ^d^	NS	NS	NS	IHC	10 y (3500 days after surgery)	Yes	No	No
Jones et al. [45]	Retrospective cohort study	USA	21	3	1	Group A (aggressive disease): 67 ^b,c^ (range: 53–81) y Group B (indolent disease) 64.5 ^b^ (range: 39–84) y	NS	G3 p53-mutated: IV: 1 (100%)	NS	IHC	4.7 ^b^ (3–9) y	No	No	No
Kang et al. [47]	Retrospective cohort study	Canada	289	26	7	64.9 ^b^ (range: 35–90)	NS	IA: 152 (69.7%) IB: 38 (17.4%) II: 5 (2.3%) IIIA: 6 (2.8%) IIIC1: 12 (5.5%) IIIC2: 4 (1.8%) IVB: 1 (0.5%)	NS	IHC	>120 mo	Yes	No	No
Kato et al. [48]	Retrospective cohort study	Japan	74	74	18	57 ^c^ (range: 37–80)	NS	I: 44 (59.5%) II: 5 (6.8%) III: 18 (24.3%) IV: 7 (9.5%)	Early-stage patients: 0 Advanced-stage patients: Platinum-containing regimens ^e^: 20 RT: 2 Refused adjuvant therapy: 3	IHC, PCR	>60 mo	Yes	No	No
Kihana et al. [21]	Retrospective cohort study	Japan	92	12	6	NS	NS	I-IV	NS	DNA sequencing	175 mo	Yes	No	No
Kitazono et al. [49]	Retrospective cohort study	Japan	82	6	2	Range: 34–88	NS	IA, IB	CT; RT	IHC	2–37 mo	No	No	No
Kobayashi et al. [50]	Retrospective cohort study	Japan	36	3	1	Range: 51–80	NS	NS	NS	IHC, NGS	NS	No	No	No
Kogata et al. [51]	Clinical Trial	Japan	344	45	12	58.0 ^b^ (SD, 11.0)	24.1 +/− 4.5	I-IV	NS	IHC, DNA sequencing	NS	No	No	No
Koshiyama et al. [18]	Retrospective cohort study	Japan	30	4	1	Range: 39–71	NS	IA	NS	IHC	NS	No	No	No
Lax et al. [52]	Retrospective cohort study	Austria	58	14	6	NS	NS	I: 39 II: 6 III: 7 IV: 2 Unknown: 4	NS	Sequencing	NS	No	No	No
Leon-Castillo et al. [53]	Retrospective cohort study	Netherlands	107	82	48	61.6 ^b^ (range: 35–87)	NS	IA: 41 (38.3%) IB: 41 (38.3%) II: 3 (2.8%) III: 16 (15%) IV: 6 (5.6%)	NS	IHC, NGS, Sanger sequencing	>5 y	Yes	No	No
Leon-Castillo et al. [35]	Retrospective cohort study	Netherlands	410	113	21	61.2 ^b^ (range: 26.7–80.5)	NS	IA: 54 (13.2%) IB: 73 (17.8%) II: 105 (25.6%) IIIA: 46 (11.2%) IIIB: 29 (7.1%) IIIC: 103 (25.1%)	RT: 200 (48.8%) CT + RT: 210 (51.2%)	IHC, NGS	6.1 ^c^ (range: 0.52–11.03) y	Yes	1.63 (1.33 to 1.99)	1.23 (1.09 to 1.39)
Li et al. [55]	Retrospective cohort study	Japan	56	5	3	p53-mutated: 55 ^c^ (range: 37–79) (n = 17)	NS	I: 40 (71.4%) II: 8 (14.3%) III: 8 (14.3%)	NS	IHC	NS	No	No	No
Li et al. [54]	Retrospective cohort study	China	70	70	10	<60 y: 60 ^d^ (85.71%) ≥60 y: 10 ^d^ (14.29%) 54.5 ^b^	NS	I/II: 43 (61.43%) III/IV: 27 (38.57%)	RT: 41 CT: 18 Adjuvant local RT: 3 Without therapy: 8	IHC, NGS, MLH1 promotor methylation testing	NS	No	No	No
Maeda et al. [56]	Retrospective cohort study	Japan	64	9	5	NS	NS	NS	NS	MSP, PCR, IHC	NS	No	No	No
Massenkeil et al. [20]	Prospective cohort study	Switzerland	23	8	3	Range: 55–82	NS	I: 4 II: 3 IV: 1	NS	DNA sequencing	NS	No	No	No
Matsumoto et al. [57]	Retrospective cohort study	USA	43	10	2	NS	NS	NS	NS	IHC, DNA sequencing	NS	No	No	No
Mazurek et al. [58]	Retrospective cohort study	Poland	55	12	6	Range: 52–74	NS	I-IV	NS	IHC	NS	No	No	No
Moreira et al. [59]	Retrospective cohort study	Portugal	230	36	10	NS	28.7	I-IV	NS	IHC	36 mo	Yes	1.97 (1.24 to 3.13)	No
Nostrand et al. [60]	Retrospective cohort study	USA	49	16	3	66 ^b^	NS	I to IV	NS	DNA sequencing	5 y	No	No	No
Nout et al. [61]	Retrospective cohort study	Netherlands	65	16	2	67.9 ^b^ (range: 51.6–84.6)	NS	I: 65 (100%)	EBRT: 31 VBT: 34	IHC, sequencing	7.3 ^c^ y	Yes	No	No
Oberndorfer et al. [62]	Retrospective cohort study	Austria	40	12	2	62.1 ^c^ (range: 29.9–83.6)	28.5 (range: 18.3–46.0)	IA: 18 (45%) IB: 12 (30%) II: 7 (17.5%) IIIC1: 2 (5.9%) IIIC2: 1 (2.5%)	NS	IHC, microsatellite instability testing	10 y (when necessary)	No	No	No
Okamoto et al. [63]	Retrospective cohort study	Japan	24	5	3	NS	NS	IA, IA, IB	NS	DNA sequencing	NS	No	No	No
Pain et al. [64]	Retrospective cohort study	Poland	291	100	10	TP53 not mutated: 67.2 ^b^ (SD, 11.8) TP53-mutated: 71.0 ^b^ (SD, 10.3)	TP53 not mutated: 28.5 TP53 mutated: 25.2	I: 174 (59.8%) II: 30 (10.3%) III: 69 (23.7%) IV: 17 (5.8%)	IVBT; EBRT; CT	IHC, NGS	>72 mo	Yes	1.66 (1.24 to 2.23)	1.29 (1.14 to 1.63)
Pijnenborg et al. [66]	Case–control study	Holand	88	17	3	70.0 ^c^ (range: 51–93)	NS	I (100%)	RT	IHC, sequencing	9–141 mo	No	No	No
Riethdorf et al. [23]	Retrospective cohort study	Germany	120	35	24	65.0 ^c^ (range: 33–99)	NS	I–IV	NS	IHC	NS	No	No	No
Rios-Doria et al. [7]	Retrospective cohort study	USA	976	87	16	Comparison <60; ≥60 y	NS	NS	RT + CT	IHC, sequencing	22.3 ^c^ mo (range: 0.5–214)	Yes	2.04 (0.48 to 8.70)	No
Sakuragi et al. [67]	Retrospective cohort study	Japan	23	3	1	59.4 ^b^ (range: 32–76)	NS	NS	NS	Yeast p53 functional assay, IHC	NS	No	No	No
Sakuragi et al. [68]	Retrospective cohort study	Japan	92	25	6	<60 y: 39 ^d^ ≥60 y: 53 ^d^	NS	I/II: 65 (70.6%) III/IV: 27 (29.4%)	RT: 49 CT: 43	IHC, PCR	58.5 ^c^ mo (n = 49) 41.0 ^c^ mo (n = 43)	Yes	No	No
Sawairiitoh et al. [19]	Retrospective cohort study	Japan	49	6	5	NS	NS	I to IV	NS	DNA sequencing	NS	No	No	No
Schultheis et al. [69]	Retrospective cohort study	USA	228	88	17	NS	NS	NS	No adjuvant therapy was used	IHC, PCR	NS	No	No	No
Shiozawa et al. [70]	Retrospective cohort study	Japan	62	9	6	NS	NS	NS	NA	IHC	NS	No	No	No
Singh et al. [71]	Retrospective cohort study	Internacional	164	32	7	NS	NS	NS	NS	IHC, DNA sequencing	NS	No	No	No
Timmerman et al. [9]	Prospective cohort study	Belgium	108	12	0	68.5 ^b^	Mean: 29.2	NS	Yes: 37 No: 71	IHC, Sanger sequencing, PCR	NS	No	No	No
Tresa et al. [72]	Retrospective cohort study	India	63	26	12	59.4 ^b^	NS	I: 31 (49.2%) II: 11 (17.5%) III: 19 (30.2%) IV: 2 (3.1%)	None: 11 (17.4%) RT: 24 (38.1%) CT only: 4 (6.3%) CT + RT: 24 (38.1%)	IHC	30.8 mo	Yes	No	No
Tsuda and Hirohashi [17]	Retrospective cohort study	Japan	52	5	3	NS	NS	I/II: (66.7%) III/IV: (33.3%)	NS	IHC, sequencing	NS	No	No	No
Vermij et al. [73]	Retrospective cohort study	Netherlands	408	112	19	61.2 ^b^ (range: 26.7–80.5)	NS	IA: 54 (13.2%) IB: 73 (17.9%) II: 105 (25.7%) III: 176 (43.1%)	Half: CT + RT/RT	IHC, NGS	>5 y	Yes	No	No
Yamauchi et al. [24]	Retrospective cohort study	Japan	35	5	3	54.5 ^b^ (range: 30- 74)	NS	NS	NS	IHC	NS	No	No	No

BMI, body mass index; CI, confidence interval; CT, chemotherapy; EBRT, external beam radiotherapy; EEC, endometrioid endometrial cancer; FIGO, International Federation of Obstetrics and Gynecology; G3, grade 3; HR, hazard ratio; IHC, immunohistochemistry; IVBT, intravaginal brachytherapy; mo, months; MSP, methylation-specific polymerase chain reaction; n, number of patients; NGS, next-generation sequencing; NS, not specified; OS, overall survival; PCR, polymerase chain reaction; PFS, progression-free survival; RT, radiotherapy; SD, standard deviation; VBT, vaginal brachytherapy; y, years. ^a^ Compared with all non-p53-mutated grade 3 endometrioid EC. ^b^ Mean value. ^c^ Median value. ^d^ Value refers to number of patients in each age category. ^e^ Including doxorubicin/cisplatin, paclitaxel/carboplatin, and cyclophosphamide/doxorubicin/cisplatin.

**Table 2 cancers-17-00038-t002:** Methodological quality assessment according to Quality in Prognostic Studies (QUIPS) tool.

Study	Study Participation	Study Attrition	Prognostic Factor Measurement	Outcome Measurement	Study Confounding	Study Analysis and Reporting	Overall
Alvarez et al. [31]	Low	Low	Low	Low	Low	Low	Low
Balestra et al. [32]	Low	Low	High	Low	Low	Low	High
Betella et al. [33]	Low	Low	Low	Low	Low	Low	Low
Bosse et al. [5]	Low	Low	Low	Low	Low	Low	Low
Brett et al. [15]	Low	Low	Low	Low	Low	Low	Low
Buchynska et al. [34]	Low	Low	Low	Low	Low	Low	Low
Chen et al. [13]	Low	Low	Low	Low	Low	Low	Low
Cuevas et al. [36]	Low	Low	Low	Low	Low	Low	Low
Da Cruz Paula et al. [65]	Low	Low	Low	Low	Low	Low	Low
Dai et al. [37]	Low	Low	Low	Low	Low	Low	Low
Daniilidou et al. [14]	Low	Low	Moderate	Low	Low	Low	Low
Dankai et al. [38]	Low	Low	Moderate	Low	Low	Low	Low
Devereaux et al. [39]	Low	Low	Moderate	Low	Low	Low	Low
Dobrzycka et al. [40]	Low	Low	Low	Low	Low	Low	Low
Enomoto et al. [22]	Low	Low	High	Low	Low	Low	High
Feng et al. [41]	Low	Low	High	Low	Low	Low	High
Hoang et al. [42]	Low	Low	Low	Low	Low	Low	Low
Huvila et al. [46]	Low	Low	Low	Low	Low	Low	Low
Huvila et al. [43]	Low	Low	Low	Low	Low	Low	Low
Inaba et al. [44]	Low	Low	Low	Low	Low	Low	Low
Jones et al. [45]	Low	Low	Low	Low	Low	Low	Low
Kang et al. [47]	Low	Low	Low	Low	Low	Low	Low
Kato et al. [48]	Low	Low	Low	Low	Low	Low	Low
Kihana et al. [21]	Low	Low	High	Low	Low	Low	High
Kitazono et al. [49]	Low	Low	Low	Low	Low	Low	Low
Kobayashi et al. [50]	Low	Low	High	Low	Low	Low	Low
Kogata et al. [51]	Low	Low	Low	Low	Low	Low	Low
Koshiyama et al. [18]	Low	Low	Low	Low	Low	Low	Low
Lax et al. [52]	Low	Low	Moderate	Low	Low	Low	Low
Leon-Castillo et al. [53]	Low	Low	Low	Low	Low	Low	Low
Leon-Castillo et al. [35]	Low	Low	Low	Low	Low	Low	Low
Li et al. [55]	Low	Low	Low	Low	Low	Low	Low
Li et al. [54]	Low	Low	Low	Low	Low	Low	Low
Maeda et al. [56]	Low	Low	High	Low	Low	Low	High
Massenkeil et al. [20]	Low	Low	High	Low	Low	Low	High
Matsumoto et al. [57]	Low	Low	Low	Low	Low	Low	Low
Mazurek et al. [58]	Low	Low	Moderate	Low	Low	Low	Low
Moreira et al. [59]	Low	Low	Moderate	Low	Low	Low	Low
Nostrand et al. [60]	Low	Low	Low	Low	Low	Low	Low
Nout et al. [61]	Low	Low	Low	Low	Low	Low	Low
Oberndorfer et al. [62]	Low	Low	Low	Low	Low	Low	Low
Okamoto et al. [63]	Low	Low	Low	Low	Low	Low	Low
Pain et al. [64]	Low	Low	Low	Low	Low	Low	Low
Pijnenborg et al. [66]	Low	Low	Low	Low	Low	Low	Low
Riethdorf et al. [23]	Low	Low	Low	Low	Low	Low	Low
Rios-Doria et al. [7]	Low	Low	High	Low	Low	Low	High
Sakuragi et al. [67]	Low	Low	Moderate	Low	Low	Low	Low
Sakuragi et al. [68]	Low	Low	Low	Low	Low	Low	Low
Sawairiitoh et al. [19]	Low	Low	Low	Low	Low	Low	Low
Schultheis et al. [69]	Low	Low	High	Low	Low	Low	High
Shiozawa et al. [70]	Low	Low	High	Low	Low	Low	High
Singh et al. [71]	Low	Low	High	Low	Low	Low	High
Timmerman et al. [9]	Low	Low	High	Low	Low	Low	High
Tresa et al. [72]	Low	Low	Low	Low	Low	Low	Low
Tsuda and Hirohashi [17]	Low	Low	Low	Low	Low	Low	Low
Vermij et al. [73]	Low	Low	Low	Low	Low	Low	Low
Yamauchi et al. [24]	Low	Low	High	Low	Low	Low	High

## Data Availability

All data relevant to this study are included in the article. Further information can be obtained from the corresponding author.

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
