# Peer review of "Abnormal p53 High-Grade Endometrioid Endometrial Cancer: A Systematic Review and Meta-Analysis"

_cancers, 2024, doi:10.3390/cancers17010038_

Round 1

Reviewer 1 Report

Comments and Suggestions for Authors

Using five databases and, following appropriate patient exclusions, this study used meta-analysis to evaluate a total of 2,528 patients with International Federation of Gynecology and Obstetrics grade 3 endometrioid endometrial cancer, harboring either mutated p53 or wild-type p53. The aim of this manuscript was to evaluate the outcome of patients with mutated p53 (30 % of the patients investigated) vs those with wild-type p53. It was found that patients harboring mutated p53 had decreased overall and progression-free survival compared with those expressing wild-type p53. It was also found that mutated p53 in endometrioid endometrial cancer is more prevalent in patients in Asia (34%) vs those in Europe (30%) and America (21%).

The studies appear to have been done vary carefully, with appropriate statistics, and the manuscript is clearly presented and well written. The results extend that which is known in the literature, and the authors are to be commended for clearly pointing out the strengths and limitations of their study (cf. the last two paragraphs on the Discussion).

There is one element that could add important additional information for this study if the authors can provide more specificity on p53. Is there sufficient information from the original papers to map mutations in the p53 gene? For example, are the mutations clustered in specific regions or spread throughout the gene?

Reviewer 2 Report

Comments and Suggestions for Authors

p53-abnormal high-grade endometrioid endometrial cancer: A systematic review and meta-analysis

This is a review manuscript which evaluated the oncological outcomes of patients with p53-abnormal FIGO grade 3 (high-grade) endometrioid endometrial cancer. The secondary objective of this study was to assess the global prevalence of abnormal p53 in grade 3 endometrial carcinomas and the geographical variations. The review found that p53-abnormal grade 3 endometrioid endometrial cancer is associated with decreased overall and progression-free survival.

The abstract is unclear. For readers of general background, I think it is challenging to parse the structure and flow of the sentences. I did not see a significance of this study based on the abstract. The last sentence in the abstract seems to be out of place.

I am really having a hard time understanding the overall goal of the manuscript. I think the authors should consider providing readers general background on the importance and significance of the study.

Overall, I think there is really no coherence in the manuscript. It failed to connect to the audience. The introduction is so short but overall very hard to parse.

The study, however, appears to be systematically performed.

There are so many things going on in the manuscript and it is hard to parse because the introduction is not clear.

The introduction shifts from cancer to carcinoma to tumor. It is hard to understand the flow because no transition words were used.

The paragraphs in the introduction did not also use transition sentences so it is hard to understand what the paper is trying to convey.

Comments on the Quality of English Language

The manuscript requires some English language copyediting.

Reviewer 3 Report

Comments and Suggestions for Authors

Manuscript ID: cancers-3318691

This review explains that endometrial cancer is the most common cancer occurring in women and is graded as 1-3, grade 3 being high-grade, aggressive, and more difficult to diagnose. Women diagnosed with p53-abnormal FIGO grade 3 endometrial cancer have an increased risk of death and cancer progression. The review highlights the outcomes of patients with p53 abnormal endometrial cancer, global prevalence, and geographical variations, showing that the estimated pooled prevalence of p-53 abnormal FIGO grade 3 endometrioid endometrial cancer is 30 % with high prevalence in Asia and is associated with decreased OS and PFS. Of the 57 studies included 6 and 8 provided OS and PFS, respectively, while 45 showed an overall minimal risk of bias. The outcomes of this review highlight that there is a huge heterogeneity in the results of this type of cancer studies and shed light on refining the molecular subtypes of grade 3 endometrial cancer which would help develop more effective treatment strategies, particularly against the most aggressive subtypes. Overall, this review provides a better understanding of grade 3 endometrial cancer and includes information on disease management in patients with this particular type of gynecologic cancer.

The review is detailed and has an interesting background. The draft is well-written, and steps/methods are described precisely and concisely which helps to understand the main concept. However, a few amendments are needed/suggested.

1-     Line-44-45, make it a single sentence, this seems that the conclusions are overextended.

2-     Line 53-54, hormone-dependent, please provide a short background on how type 1 is hormone-driven and how type II is not.

3-     Line 90, capitalize the letter “t”

4-     Line-93-94, stick to the round brackets, please keep it consistent in the draft.

5-     Line 111, can any estimated reproductive age of women be added/mentioned in this sentence?

6-     Discussion, the line “Bearing this is mind, in G3 endometrioid endometrial carcinoma with mutated TP53, aberrant p53 signaling pathways contribute to uncontrolled cell proliferation, genomic instability, and resistance to apoptosis” does not sound correct. Please correct.

7-     Discussion, Line “After the results of this review, one can speculate that in the presence of p53-abnormal G3 endometrioid endometrial carcinoma, patients may require differing therapeutic options, according with the stage of disease at presentation.”, It is suggested to modify and correct the sentence.

8-     Discussion, paragraph no. 4, line no. 4, please remove the brackets.

9-     Discussion, paragraph no. 4, line no. 4-6, what is the age of diagnosis? Please highlight.

10- Discussion, paragraph no. 4, line no. 12, “bearing in mind”, is the repetition, please substitute these words.

11- In the discussion part, the last two paragraphs, explain the strengths and limitations of this review, however, it is suggested to adjust these two paragraphs under the title “Strengths and limitations” and then explain them in a few sentences so that they are understandable to the reader.

12- In the conclusions section, the very last sentence does not seem correct/ well explained. It is suggested that it be reorganized for a better understanding. 

Round 2

Reviewer 2 Report

Comments and Suggestions for Authors

The manuscript has improved and answered the comments.